# A T-cell receptor escape channel allows broad T-cell response to CD1b and membrane phospholipids

Adam Shahine[1,2], Peter Reinink [3,4], Josephine F. Reijneveld[3,4,5], Stephanie Gras[1,2], Mira Holzheimer[5], Tan-Yun Cheng[4], Adriaan J. Minnaard[5], John D. Altman [6], Steffi Lenz[3], Jacques Prandi[7], Joanna Kubler-Kielb[8], D. Branch Moody [4], Jamie Rossjohn [1,2,9] & Ildiko Van Rhijn [3,4]

CD1 proteins are expressed on dendritic cells, where they display lipid antigens to T-cell receptors (TCRs). Here we describe T-cell autoreactivity towards ubiquitous human membrane phospholipids presented by CD1b. These T-cells discriminate between two major types of lipids, sphingolipids and phospholipids, but were broadly cross-reactive towards diverse phospholipids including phosphatidylcholine, phosphatidylinositol and phosphatidylethanolamine. The crystal structure of a representative TCR bound to CD1b-phosphatidylcholine provides a molecular mechanism for this promiscuous recognition. We observe a lateral escape channel in the TCR, which shunted phospholipid head groups sideways along the CD1b-TCR interface, without contacting the TCR. Instead the TCR recognition site involved the neck region phosphate that is common to all major self-phospholipids but absent in sphingolipids. Whereas prior studies have focused on foreign lipids or rare self-lipids, we define a new molecular mechanism of promiscuous recognition of common self-phospholipids including those that are known targets in human autoimmune disease.

[1] Infection and Immunity Program and Department of Biochemistry and Molecular Biology, Biomedicine Discovery Institute, Monash University, Clayton, VIC 3800, Australia. [2] Australian Research Council Centre of Excellence in Advanced Molecular Imaging, Monash University, Clayton, VIC 3800, Australia. [3] Department of Infectious Diseases and Immunology, Faculty of Veterinary Medicine, Utrecht University, Yalelaan 1, 3584CL Utrecht, The Netherlands. [4] Brigham and Women's Hospital Division of Rheumatology, Immunology and Allergy and Harvard Medical School, Boston, MA 02115, USA. [5] Stratingh Institute for Chemistry, University of Groningen, 9747AG Groningen, The Netherlands. [6] Department of Microbiology and Immunology, Emory University School of Medicine, 1510 Clifton Road, Atlanta, GA 30322, USA. [7] Institut de Pharmacologie et de Biologie Structurale, Université de Toulouse, CNRS, Université Paul Sabatier, 31077 Toulouse, France. [8] National Institute of Child Health and Human Development, National Institutes of Health, 9000 Rockville Pike, Bethesda, MD 20892, USA. [9] Institute of Infection and Immunity, School of Medicine, Cardiff University, Heath Park, Cardiff CF14 4XN, UK. Correspondence and requests for materials should be addressed to J.R. (email: Jamie.rossjohn@monash.edu) or to I.V.R. (email: I.vanRhijn@uu.nl)

Human αβ T-cells recognize antigen complexes formed from MHC proteins bound to diverse peptides, CD1 proteins bound to diverse lipids, and MR1 presenting small molecules[1]. When an αβ T-cell receptor (TCR) recognizes a self-peptide presented by an MHC protein, the sequence and structure of the peptide bound controls whether the T-cells respond[2,3]. T-cells with high affinity for MHC–self-peptide are rarely isolated in the periphery because the negative selection mechanism in the thymus prevents most MHC-restricted auto-reactive T-cells from entering the circulation. Nevertheless, T-cell autoreactivity towards self-peptide–MHC complexes occurs, which can manifest as autoimmune disease[4]. With increasing evidence that CD1 proteins present highly diverse self and foreign lipids to αβ T-cells[3], the interrelated questions of negative selection, T-cell fine specificity for lipids and ratios of self and foreign reactive T-cells in the periphery are likewise coming to the fore for lipids.

In this study we measured T-cell responses to lipid antigens using CD1b tetramers, which bind to antigen-specific TCRs[5]. Although the four types of human CD1 antigen-presenting molecules (CD1a, CD1b, CD1c, CD1d) are related in structure[3], CD1b has distinct cellular and immunological functions. As compared to other human CD1 proteins, CD1b shows particularly strong recycling through the most acidic endosomes[6,7], use of a particularly capacious cleft to capture large lipids[8], and it is uniquely expressed on a subset of activated macrophages in the periphery[9]. CD1b was the first of the human CD1 proteins discovered to present exogenous antigens[10]. There is extensive evidence for its presentation of mycobacterial lipids to T-cells, including mycolic acids[11], lipoarabinomannan[12], glucose monomycolate[13], phosphatidylinositol dimannoside (PIM2)[14], glycerol monomycolate[15] and sulfoglycolipids[16,17]. More recent results show that CD1 protein expression[18] or CD1-restricted T-cells, can respond to lipids from other human pathogens, including Borrelia burgdorferi[19] and Gram negative bacteria[20].

To investigate potentially new targets of CD1b-mediated T-cell response, we undertook an effort to detect human T-cells responding to mycobacterial diacyltrehalose, Borrelia burgdorferi glycolipid 2 and bacterial lipid extracts. The goal of the study was to select for foreign antigen-specific T-cells, and in so doing, identify new bacterial antigens. The approach was based on the premise that mechanisms of negative selection likely bias the peripheral T-cell repertoire toward TCRs with specificity for foreign lipids. We selected for foreign lipid reactive T-cells by using CD1b tetramers treated with pure bacterial ligands or complex lipid extracts from bacterial pathogens. Unexpectedly, we derived a series of CD1b-autoreactive T-cells lines with broad responses to common self-phospholipids that are widely expressed in human cellular membranes. After identifying a distinct pattern whereby T-cells respond broadly to phospholipids in preference to sphingolipids, we solve the basis of this response via a ternary crystal structure of a CD1b–phosphatidylcholine–TCR complex.

## Results

**CD1b-autoreactive T-cell lines are frequently isolated**. To isolate bacteria reactive T-cells, we loaded CD1b tetramers with foreign lipid antigens or lipid extract (Table 1). After several rounds of T-cell sorting and expansion from blood bank-derived buffy coats (BC) or peripheral blood mononuclear cells (PBMC) from healthy donors (HD), we obtained oligoclonal T-cell populations that were >85% CD1b tetramer+ (Fig. 1a and Supplementary Figs. 1–5). Prior approaches using activation assays largely failed to detect CD1b-reactive cells in unfractionated PBMCs[21,22], but these tetramer-based enrichment procedures

**Table 1 T-cell lines and subpopulations**

| Human subject | Selecting CD1b tetramer | Name of T-cell subpopulation | CD1b tetramer staining |
|---|---|---|---|
| BC10 | BBGL1 | BC10A | PL brighter than SL |
| BC8 | Brucella lipid extract | BC8A | LPA |
| | | BC8B | PL but not SL |
| BC24 | BBGL2 | BC24A | BBGL2 |
| | | BC24B | PL but not SL |
| | | BC24C | PG |
| HD1 | Diacyltrehalose | HD1A | Diacyltrehalose |
| | | HD1B | PL but not SL |
| BC13 | Salmonella lipid extract | BC13A | PL and SL |

Summary of the T-cell lines used, including the human subject number from which the original blood PBMC were derived, the tetramer that was used to isolate the T-cells, the antigen, the names of the subpopulations, and their reactivity pattern. Subject numbers preceded by BC of HD refer to blood bank-derived buffy coats or healthy donors, respectively. *PL* phospholipids, *SL* sphingolipids, *LPA* lysophosphatidic acid, *PG* phosphatidylglycerol

succeeded in recovering CD1b tetramer-positive T-cells in each of five unrelated donors with a measured precursor frequency of about 1 in 10⁴ T-cells, ranging between 0.018 and 0.33% (Supplementary Figs. 1–5 and Table 1). These numbers are somewhat higher than naive MHC-restricted T-cells and lower than human NKT-cells[23]. The initial CD1b tetramer+ cell populations (HD1, BC8, BC10, BC13, BC24) were named according to the donor, and sub-lines A, B, or C were designated when the line could be sorted into clearly distinct subpopulations (Supplementary Figs. 1–5).

As expected, two cell lines recognized bacterial lipids: BC24A and HD1A (Supplementary Figs. 3-4 and Table 1). However, all other lines showed functional autoreactivity to CD1b proteins in the absence of added foreign lipid. For example, lines HD1B, BC24, BC10A, as well as line BC8 and two lines derived thereof, BC8A and BC8B, all showed interferon-γ release in response to CD1b-transfected but not CD1a- or CD1c-transfected human (K562 or C1R) cells (Fig. 1b). Similarly, line BC13A upregulates the activation marker in response to CD1b-transfected C1R cells (Fig. 1c). Further, CD1b-dependent autoreactive T-cell responses could be seen against monocyte-derived dendritic cells, which represent physiological APCs (Supplementary Fig. 6). Thus, despite using methods to select foreign lipid reactive T-cells, all new lines showed substantial autoreactivity to CD1b.

Although somewhat counterintuitive, the recovery of CD1b-autoreactive T-cells has been observed previously[20] and could be explained by extremely cross-reactive TCRs that do not discriminate lipid structure, retention of self-lipids from the expression system used to make CD1b tetramers, or antigenic lipids expressed both by mammalian cells and bacteria. This functional autoreactivity observed in activation assays was confirmed using untreated CD1b tetramers that carried endogenous self-lipids from the human expression system (CD1b-endo) and CD1a-endo as a negative control (Fig. 1d). For five CD1b-autoreactive sub-lines (HD1B, BC10A, BC13A, BC8B, BC8A), the pattern of CD1b-specific tetramer staining matched the observed CD1b-specific functional response. For donor BC24, functional reactivity to CD1b (Fig. 1b) was likely explained by the response of the sub-line BC24B rather than BC24C (Fig. 1d). Overall, these results confirmed CD1b autoreactivity using a separate method that relies on TCR binding to CD1b-endo complexes.

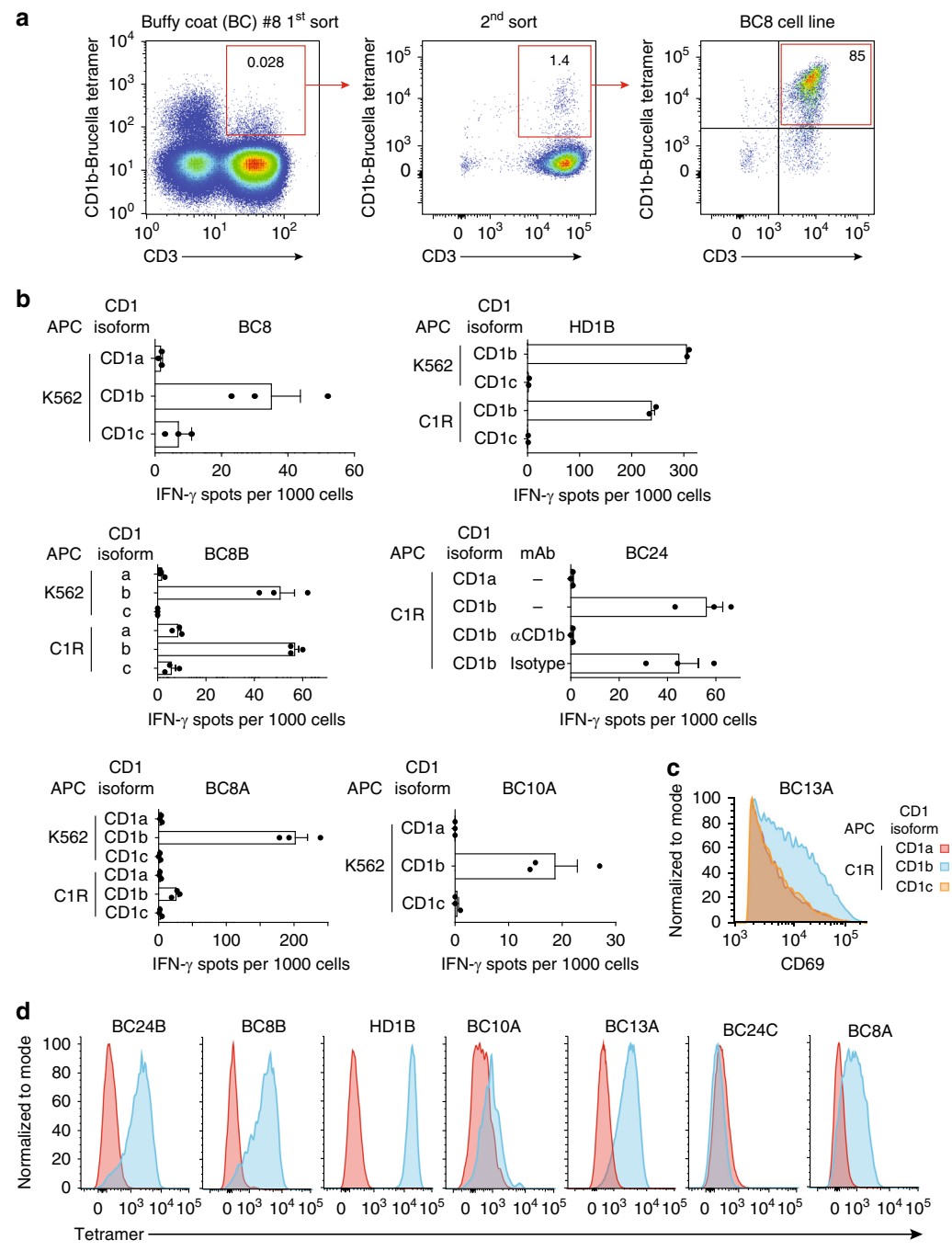

**Fig. 1** CD1b tetramer selection generates autoreactive T-cell lines. **a** Peripheral blood mononuclear cells from blood bank-derived buffy coat 8 (BC8) were sorted based on expression of CD3 and binding to CD1b tetramers treated with *Brucella* lipid extract, followed by expansion in vitro. Numbers in the outlined red areas indicate percent cells in gate. The generation of additional T-cell lines is shown in Supplementary Figures 1–5. **b** IFN-γ ELISPOT assay of T-cell lines and subpopulations stimulated with K562 or C1R cells transfected to express CD1a, CD1b, or CD1c in the absence of exogenously added antigen. Error bars represent standard error of the mean (SEM) of triplicate wells. **c** Jurkat cells transduced to express the BC13A TCR were stimulated with C1R cells transfected to express CD1a, CD1b, CD1c in the absence of exogenously added antigen. After 24 h, CD69 expression was determined by flow cytometry. **d** Polyclonal T-cell populations from the indicated donor were stained with CD1a-endo or CD1b-endo tetramers. T-cell line derivation (**a**) and screening (**b**, **d**) was performed once on each of five donors. The experiments in **c** were performed twice with comparable results. APC antigen-presenting cell

**Self-lipids in CD1b complexes**. Because few chemically identified self-antigens are known for the human CD1b system, we used these CD1b-endo complexes and the panel of CD1b-autoreactive lines as a route to discover lipid autoantigens. Using a recently reported mass spectrometry approach[24], we identified several lipids present in the CD1b-endo monomers used to make CD1b tetramers. After monomers were treated with chloroform, methanol and water, we captured lipids in the organic phase and precipitated proteins into the aqueous phase. Lipid eluents were subjected to negative mode nanoelectrospray ionization

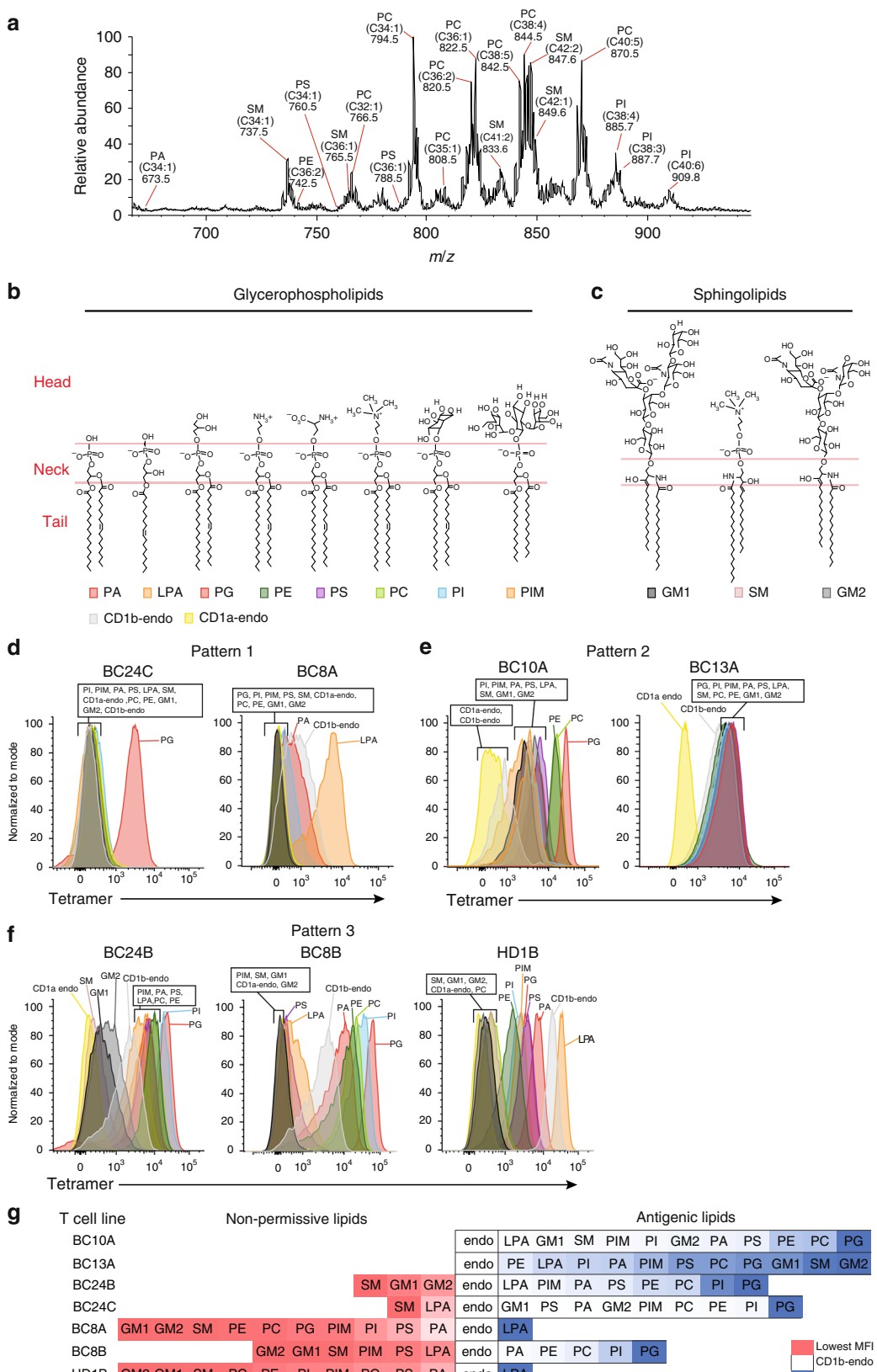

(nanoESI) ion trapping MS, identifying ions corresponding to deprotonated ions of phosphatidic acid (PA, *m/z* 673.5), phosphatidylethanolamine (PE, *m/z* 742.5), phosphatidylserine (PS, *m/z* 760.5), and phosphaditylinositol (PI, *m/z* 885.7, 887.7, 909.8). Further we detected chloride adducts of phosphatidylcholine (PC,

*m/z* 794.5, 808.5, 820.5, 822.5, 842.5 844.5) and sphingomyelin (SM, *m/z* 737.5 765.5, 849.6). For some ions, these tentative identifications based on mass alone could not be formalized due to limited MS intensity or mass accuracy (Fig. 2a). However, for all six lipid classes, CID mass spectrometry identified diagnostic

**Fig. 2** Patterns of antigen specificity among CD1b-restricted T-cell lines. **a** Lipids were eluted from untreated CD1b proteins and analyzed by mass spectrometry. The identities of the indicated lipids were initially established by based on *m/z* value matching to masses of known compounds. All six lipid classes were confirmed by collisional mass spectrometry based on diagnostic fragmentation patterns (Supplementary Fig. 7). **b–f** A panel consisting of 12 allophycocyanin-labeled CD1b tetramers and one CD1a tetramer, treated with the indicated phospholipid (**b**) or sphingolipid (**c**) or mock treated (endo), was used to stain the indicated T-cell lines. **g** Tetramer binding data are arranged according to mean fluorescence intensity (MFI) and aligned relative to the MFI of CD1b-endo staining. An experiment using the full panel of 13 tetramers and all T-cell lines was performed once. Replicate experiments with a smaller subset of the tetramer panel were conducted at least once for each T-cell line and tetramer

ions leading to their confirmed structural identification (Supplementary Fig. 7). For example, cleavage of PA[25] *m/z* 673.5 generated fragments identifiable as fatty acids (*m/z* 255, 281), and monoacylphosphoglycerols (*m/z* 391, 409, 417) (Supplementary Fig. 7a). Considering the combined length of acylglycerol or sphingosine units, the lipid anchors of ligands ranged from C30-C42, with strongest signals seen for C38 and C40 lipids (Fig. 2a). Similar results were obtained using HPLC–TOF-MS methods (Supplementary Fig. 8).

**Discovery of self-lipid antigens**. In addition to lipids directly identified in CD1b-endo complexes, we designed a larger panel of lipids for screening in CD1b tetramer assays. These additional lipids included lysophosphatidic acid (LPA), ganglioside M1 (GM1) and GM2 and synthetic phosphatidylinositol dimannoside (PIM2) (Fig. 2b, c). These lipids were chosen because they are naturally occurring, available in pure forms, and were known as CD1 ligands in other contexts[14,26,27]. Recent studies on human CD1a[28,29] and CD1c[30] have shown how lipids are more likely to act as autoantigens if they have small head groups or fewer lipid tails[20,24,31,32]. Smaller CD1 ligands can remain sequestered within CD1 clefts, where head groups do not interfere with CD1-TCR contacts. Two decades after the discovery of neural tissue-derived glycosphingolipid presentation by CD1b[33], ubiquitous antigen-presenting cell-intrinsic CD1b-presented autoantigens have only been recently discovered[20,31] and general size constraints are unknown. Accordingly, lipids in this panel spanned a large range of head group size, from PA, which has no head group, to lipids with small head groups (PS, PE, PC, SM) up to glycolipids with one polyalcohol (inositol), or three (PIM2), four (GM2) or five (GM1) carbohydrates. Also, this panel sampled the two main types of lipids in mammalian membranes: glycerophospholipids and sphingolipids. Glycerophospholipids differ from sphingolipids in their 'neck region,' where glycerophospholipids contain an anionic phosphate ester, and sphingolipids have an amide linkage (Fig. 2b, c). Thus, in addition to acting as probes to define new antigenic molecules, this panel tested lipid tails, neck regions and head groups as determinants of CD1b-TCR contact.

We treated CD1b-endo tetramers with vehicle alone and separately with each of the 11 lipids. These antigen-loaded CD1b tetramers were screened for staining of 7 CD1b-autoreactive T-cell populations. Tetramers avoid certain false positive results seen in activation assays because tetramers rely on direct physical interactions of TCRs with antigen complexes, but tetramers also give false positive or negative results in some circumstances. Therefore, it is notable that for every lipid tested, alternatively loaded tetramers generated differential upregulating or down-regulated binding, based on the T-cell line used (Fig. 2d–f). Visualization of the data as a matrix of 77 staining conditions makes clear that the seven lines provided positive and negative controls for one another (Fig. 2g). No lipid generally failed to bind tetramers, generally inactivated tetramers or generally caused tetramers to bind non-specifically to T-cells. Instead differing patterns within the matrix were most consistent with differential interactions with clonally distributed TCRs and the type of lipid loaded onto CD1b. This matrix identified many

lipids that could block (non-permissive ligands) or enhance (antigens) CD1b-endo staining of T-cells, demonstrating broad recognition of self-lipids by human CD1b-reactive T-cells.

**Three patterns of self-antigen reactivity**. Further, by observing larger patterns in which few or many lipids affected T-cell staining, as well as the general classes of lipids that controlled staining, we could identify three general patterns of T-cell specificity for CD1b lipid-complexes. Pattern 1 was observed for alternatively loaded tetramer staining of BC24C and BC8A T-cell populations. Here T-cells recognized phospholipids in preference to sphingolipids, and they specifically stained with phospholipids carrying small head groups, such as PG and LPA (Fig. 2d). The staining was highly specific for the lipid loaded, as there was no cross-reactive recognition of membrane phospholipids with larger head groups such as PC and PI. These results confirmed a known antigen response pattern previously observed in clones PG90 and PG10[20,24] and broadened the result to additional human donors.

In addition, this matrix identified two previously unknown lipid antigen reactivity patterns for CD1b. Two T-cell populations, BC10A and BC13A, constituted pattern 2. They recognized CD1b tetramers treated with many phospholipids or sphingolipids (Fig. 2e). Despite this extreme promiscuity for lipid structure, both lines were still specific for the CD1b isoform in T-cell activation (Fig. 1) and tetramer staining (Fig. 2e) assays. Last, pattern 3 was exemplified by BC24B, BC8B, and HD1B (Fig. 2f). Here T-cells showed broadly cross-reactive recognition of many classes of self-phospholipids, including PG, PA, PI, PC, PS, and PE. Unlike pattern 1, binding was highly promiscuous among phospholipid types, with T-cells cross-reacting with phospholipids lacking a head group (PA) and those with varying head group size and structure. Like pattern 1, these lines were not cross-reactive to any tested sphingolipid, including gangliosides GM1, GM2, and SM (Fig. 2f). Of note, the phospholipid PC and the sphingolipid SM have the same phosphocholine head group (Fig. 2b) but are differentially recognized. Thus, pattern 3 suggests that the T-cell response might be in some way specific for the phosphoglycerol neck region that differentiates sphingolipids from phospholipids.

Thus, screening with major classes of cellular self-lipids was highly informative regarding the identity of individual antigens identified, including PS, PE, PI, and SM. Pattern 3, with apparent 'neck region' specificity that translated into preference for phospholipids over sphingolipids, was particularly intriguing. To our knowledge, this pattern lacked precedent. Yet, it could potentially explain how T-cells respond differentially to the two major classes of membrane lipids present in human cells. Also from a structural perspective, pattern 3 was difficult to understand. The only prior structural study of an autoreactive TCR bound to CD1b, CD1b-PG-PG90 TCR, shows that head group protrudes 'upward' to contact the TCR[24]. If head groups usually dominate TCR contact with phospholipids, it was unclear how pattern 3 TCRs could somehow equally recognize large (PI, PC), small (PS, PE, PG), or absent (PA) head groups, yet still discriminate neck region structures, which are likely positioned further from the TCR and nearer to CD1b. Therefore, we chose

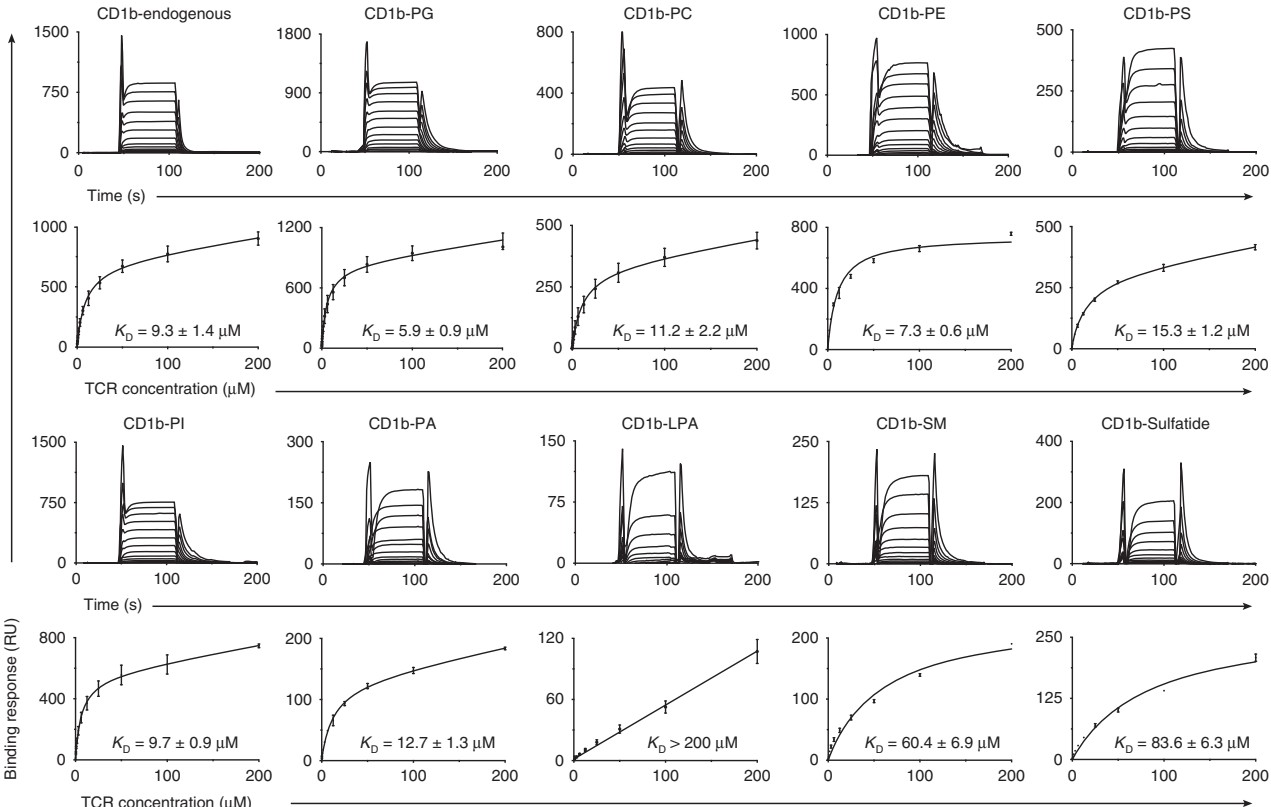

**Fig. 3** Steady-state affinity measurements of BC8B TCR against CD1b by SPR. Steady-state affinity measurements by SPR of the BC8B TCR against CD1b presenting a range of phospholipid and non-phospholipid antigens. Three (CD1b-endo, CD1b-PS, and CD1b-PC) or two independent (all other antigens) SPR experiments were conducted. Steady-state ($K_D$) values and error bars represent mean + SEM in μM of duplicate measurements. Sensograms (upper) and equilibrium curves (lower) were generated in Graphpad Prism 7.0

the BC8B TCR (Fig. 2f) as a representative pattern 3 T-cell for detailed structural studies.

**Binding of a broadly cross-reactive TCR to CD1b-phospholipid.** First, we cloned and sequenced the TCR of BC8B. The TCR α chain complementarity determining region 3 (CDR3α) sequence CALTPSGGYQKVTF was formed by the gene segments TRAV9-2 and TRAJ13, and a CDR3β sequence CASSMPGLRSSYEQYF was formed by TRBV6-2 and TRBJ2-7. TCR heterodimers were generated and purified according to published protocols[24]. In surface plasmon resonance (SPR) assays, CD1b was loaded with various lipid antigens, coupled to the surface, and recombinant BC8B TCR was the analyte (Fig. 3). For CD1b loaded with PC, PE, PA, PI, PG, or PS, steady-state affinities were measured in the range of 5.9–15.3 μM, which represents a high affinity interaction for any self-ligand[2]. Very weak binding was detected to CD1b carrying the single chain lipid LPA ($K_D > 200$ μM), consistent with the absence of staining with LPA-treated tetramers (Fig. 2f, Supplementary Fig. 2). The BC8B TCR detectably interacted with CD1b presenting the sphingolipid antigens, SM and sulfatide, but did so with extremely low affinity, 60.4 and 83.6 μM, respectively, as compared with CD1b-phospholipids. Although there were some differences in avidity or affinity among individual phospholipids, both tetramer and SPR-based assays showed a clear preference for self-phospholipids over sphingolipids and promiscuity among phospholipids, implying neck region specificity.

**BC8B TCR docking onto CD1b-phospholipid.** Next, we determined the crystal structure of the CD1b-PC–BC8B TCR complex to a resolution of 2.4 Å (Supplementary Table 1, Fig. 4a). The

entire PC molecule was clearly defined in the electron density map (Supplementary Fig. 9), so that its interaction with the TCR could be analyzed in detail. Similar to prior CD1b complexes with self-lipids[24,34,35] and the detection of CD1b ligands whose lipids are smaller than the CD1b cleft (Fig. 2a, Supplementary Fig. 9), the large cleft of CD1b contained a scaffold lipid in addition to the acyl chains of PC (Supplementary Fig. 9). The TCR docked centrally with an approximate 110° angle across the antigen-binding cleft, where the TCR contacted the lipid and CD1b (Fig. 4a, b). The total buried surface area (BSA) of the interaction was 1990 Å², with equivalent contributions by the TCR α (53%) and β (47%) chain (Fig. 4b).

The TCR α-chain dominated contacts with PC, where the CDR1α and CDR3α regions (20% and 30% BSA, respectively) interact with the phosphate, sn1 acyl tail, and both the α1 and α2 helices CD1b (Supplementary Table 2, Fig. 4b, c). Specifically, the main chain of Y36α and P37α in the CDR1α loop and P108α in the CDR3α loop form hydrogen bonds with the phosphate. S109α of the CDR3α loop hydrogen bonds with the hydroxyl group on the sn1 lipid tail (Fig. 4c). The hydroxyl group of Y36α of the CDR1α forms a hydrogen bond with S109α in the CDR3α, aiding in stabilizing the TCR interaction with PC. Furthermore, Y36α and K58α of the CDR1α and CDR2α loops, respectively, make contacts proximally to the CD1b α2 hinge regions (Supplementary Table 2, Supplementary Fig. 10a). Whereas CDR3α loop contacts the PC neck region and sn1 tail regions (Fig. 4c), Y113α contacts R79 and E80 of the α1 helix of CD1b to further anchor CD1b-PC docking (Supplementary Table 2, Supplementary Fig. 10b).

In contrast, TCR β lacks interactions with PC, and instead contacts the F′ portal of CD1b (32% BSA) (Fig. 4b,

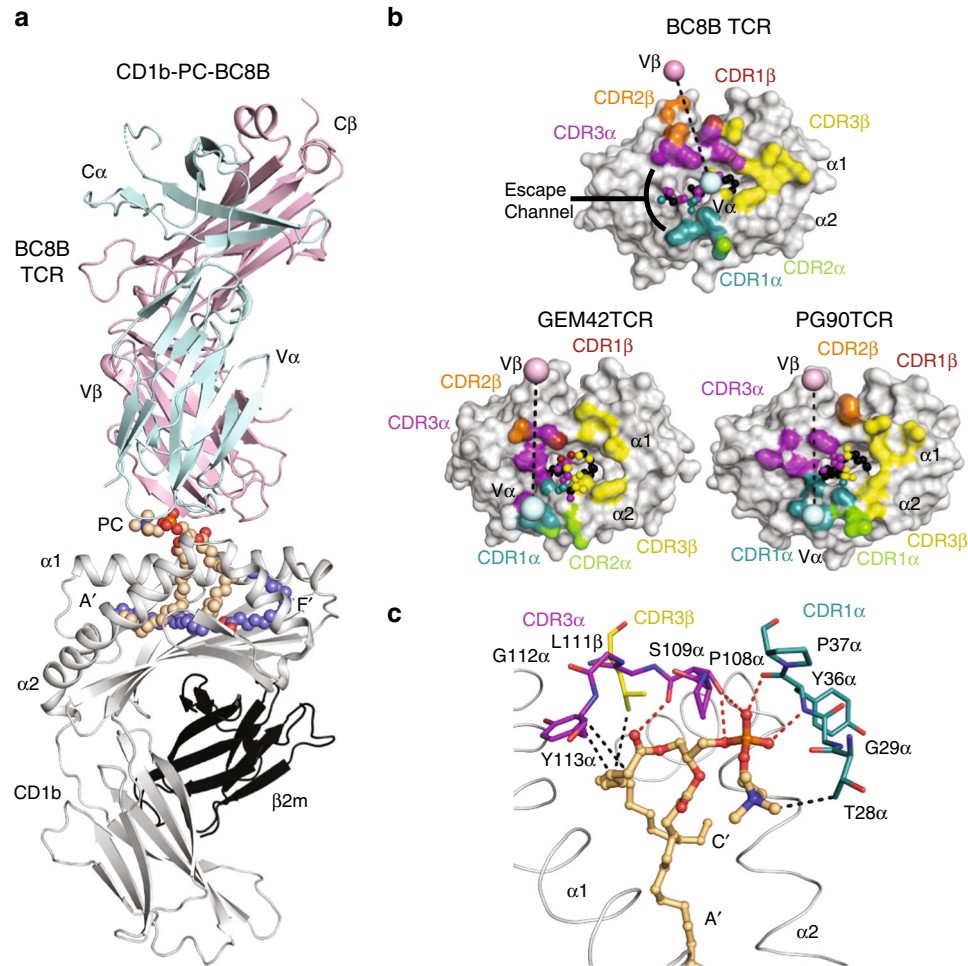

**Fig. 4** Overview of the CD1b-PC-BC8B crystal structure. **a** Overview of crystal structure of CD1b presenting PC in complex with the BC8B TCR. CD1b (gray), β2 m (black), BC8B TCR α chain (light blue), and β chain (light pink) are represented as ribbons, with PC (light orange) and the scaffold lipid (dark blue) represented as spheres. Dashed lines indicate missing residues in the structure due to lack of electron density. **b** Footprint of BC8B CDR loops on CD1b-PC, in comparison to the PG90 TCR and GEM42 TCR CDR loops onto CD1b-PG and CD1b-GMM, respectively. CD1b and antigens are represented as gray surface and spheres, respectively, with points of contacts with TCR CDRs indicated as follows: CDR1α (teal), CDR2α (green), CDR3α (purple), CDR1β (red), CDR2β (orange), and CDR3β (yellow). Ligand atoms not contacted by the BC8B TCR are indicated in black. TCR α and β centers of mass are indicated by spheres and colored as indicated in **a**. Escape channel formed by BC8B TCR CDR1α and CDR3α regions is labeled. **c** Contacts between the BC8B TCR CDR1α (teal), CDR3α (purple), and CDR3β (yellow) regions, CD1b (gray), and the PC antigen (light orange). Amino acid residues involved in contacts are represented as sticks, with the CDR regions, and α1 and α2 helices of CD1b represented as ribbons. Hydrogen bonds and hydrophobic interactions are indicated as red and black dashes, respectively. Oxygen, nitrogen, and phosphate ions are colored red, blue, and orange, respectively

Supplementary Table 2, Supplementary Fig. 10c-d). CDR3β docks across the α1 helix, with the sulfur side chain and hydroxyl backbone of M108β and P109β, respectively, bonding with R79 (Supplementary Fig. 10d). In fact, the CDR3β loop penetrates the F′ portal to gain access to the CD1b cleft, where L111β contacts the buried L154 on the α2 helix and L111β amine backbone and E80 on the α1 helix (Supplementary Fig. 10d). TCR docking onto CD1b is further solidified by R112.1β forming salt bridges and a hydrogen bond with D87 and D83, respectively (Supplementary Figs. 10d and 11). While the CDR1β and CDR2β loops have lesser roles, hydrogen bond formation between Y31β and Y113α of the CDR1β and CDR3α loops, respectively (Supplementary Fig. 10c), stabilize the PC antigen-binding region architecture. Overall, the BC8B TCR binds near the center of CD1b with both chains providing defined contacts that explain CD1b and PC specificity.

**Conserved TCR docking modes on CD1b**. Comparison of the BC8B TCR with the GEM42 and PG90 TCRs, the only ternary TCR structures currently solved with CD1b[24,36], provides insight

into key commonalities that underpin antigen recognition (Fig. 4b, Supplementary Fig. 11). All three TCRs exhibit a β chain center of mass above the α1 helix of CD1b, with contacts spreading across the α1 helix, and bridging towards the α2 helix (Fig. 4b)[24,36]. The CDR3β region contributes to significant contacts with CD1b in each case. TCR stabilization on CD1b is partially driven by salt bridge formation between key acidic residues on the α1 helix of CD1b, and a non-germline encoded arginine residue present on the N-region of each CDR3β loop. Specifically, R112.1β of the BC8B TCR binds D87 of CD1b (Supplementary Figs. 9d and 11), whereas salt bridges form between R110β of the PG90 TCR and E80 and D83, respectively, and R109β of GEM42 TCR binds E80 of CD1b (Supplementary Fig. 11). These polar interactions contribute to anchoring the respective TCRs to allow for optimal lipid antigen co-recognition. These similarities are observed despite the lack of significant conservation of TCR gene usages[24,36]. Conversely, both the BC8B and GEM42 TCRs share TRBV6-2 genes, but their respective CD1-TCR structures reveal the CDR1β and CDR2β regions have minimal roles in CD1b and antigen contacts (Fig. 4b,

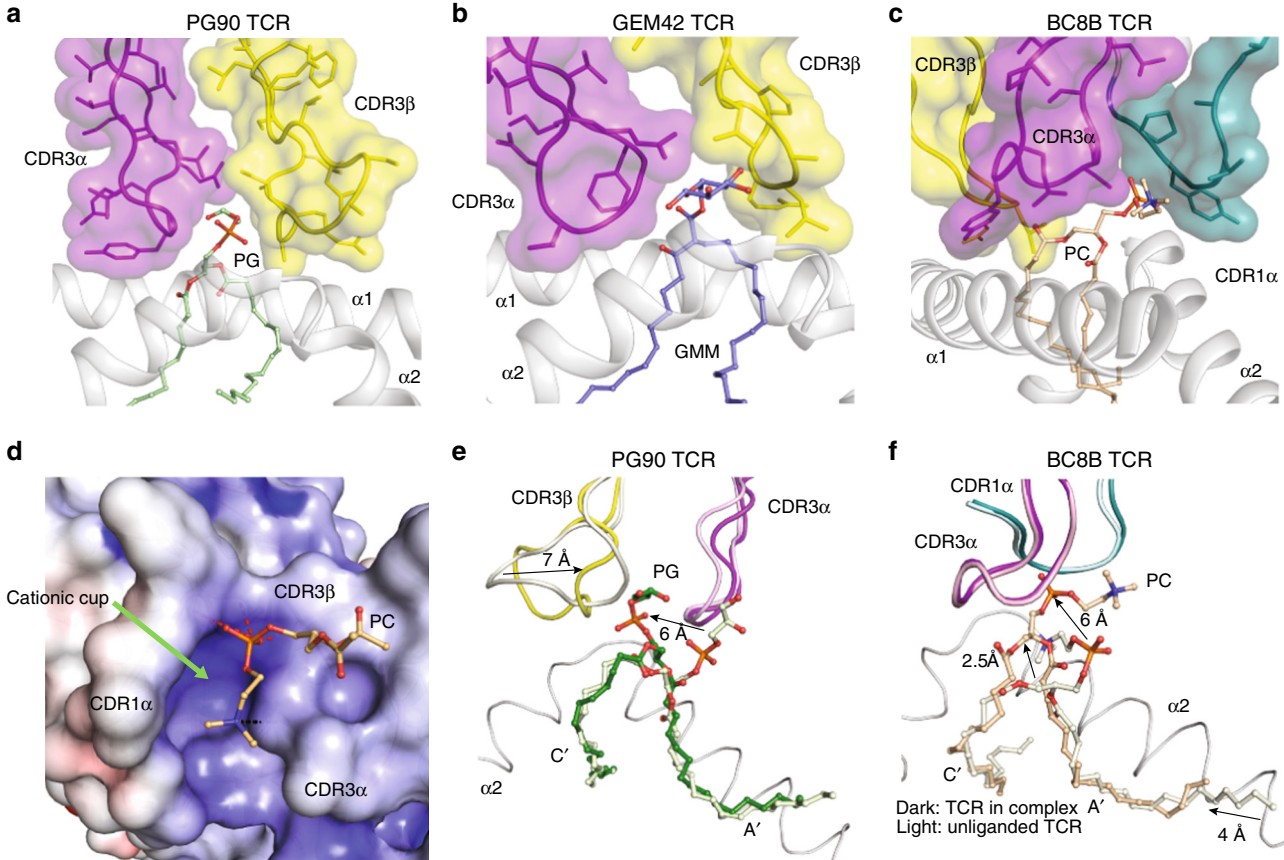

**Fig. 5** Cationic cup formation and rearrangement upon TCR docking. **a–c** CDR loop contacts with antigen head group. Surface representation of the CDR1α (teal), CDR3α (purple), and CDR3β (yellow) loops of **a** the PG90 TCR in contact with PG (green), **b** the GEM42 TCR in contact with GMM (blue), and **c** the BC8B TCR in contact with PC (light orange). **d** Electrostatic potential of the cationic cup (green arrow) formed by the BC8B TCR viewed bottom up from CD1b to the TCR. Hydrogen bonds and hydrophobic interactions between residues within the cationic cup and PC are indicated in red and black dashes, respectively. The potential contours are shown on a scale from +10.0 (positive charge, blue) to −10.0 $k_BT\,e^{-1}$ (negative charge, red); white indicates a value close to 0 $k_BT\,e^{-1}$ (neutral charge). Cationic cup and direction of lateral escape are indicated by green and black arrows, respectively. **b** Cationic cup and lipid antigen remodeling upon docking of **e** the PG90 TCR, and **f** the BC8B TCR. CDR loops of the unliganded TCR and TCR in complex with CD1b, as well as lipid antigens presented in CD1b binary and in complex with TCR structures, are represented in lighter and darker colors, respectively, as per Fig. 4a. Directions of movement are indicated by black arrows

Supplementary Table 2). Instead, the CDR1β region has an indirect role in stabilizing the antigen-binding architecture of the CDR3α and CDR3β regions of the BC8B and GEM42 TCRs, respectively (Supplementary Fig. 10c). Considering the contacts between the TCRs and their respective lipid antigens, all three TCR footprints span across the F′ portal, allowing contact with elements of the glycolipid (GEM42) or phospholipid (BC8B and PG90) that protrude through the F′ portal onto the surface of CD1b (Figs. 4b and 5a–c).

**A TCR escape channel limits antigen contact**. Despite these commonalities, the key difference in antigen recognition is that the PG90 and GEM42 TCRs are highly specific for antigen head groups, whereas the BC8 TCR is not. For the GEM42 and PG90 TCRs, the TCRs make a tight seal on CD1b, forming a gasket-like, O-ring seal that fully surrounds the F′ portal and protruding antigen (Figs. 4b and 5a, b). Surrounded on all sides, glucose or phospholipid head groups are forced upward towards the interface of the TCR α and β chains. Here the two chains act like two arms of tweezers that extensively surround and broadly contact head groups[24,36] (Fig. 5a, b). In contrast, the contacts between the BC8B TCR and lipid antigen are predominantly limited to the TCR α-chain, where the CDR1α and CDR3α loops contact both the phosphocholine neck region and the sn1 lipid tail (Fig. 4c and

5c). This alternative binding mechanism propagates a differing docking angle and TCR-CD1b contact interface creating a reversed C-shaped TCR contact footprint (Figs. 4b and 5d).

This forms a wide channel between the CDR1α and CDR3α loops that allows the choline head group of PC to escape laterally across the surface of CD1b, where it rests on the A′ roof of CD1b (Figs. 4b and 6a). The 'bent' conformation of the antigen allows the anionic phosphate unit of the neck region to cradle within a cationic cup-like surface in the TCR (Fig. 5d), with the choline group escaping TCR contact (Fig. 4c). Overall, the BC8B TCR escape channel provides a straightforward explanation for both the neck region specificity for phosphoglycerol lipids over sphingolipids, as well as the lack of head group specificity for phospholipid subclasses: the head groups protrude laterally through a channel in the TCR.

**Two mechanisms of cationic cup formation**. The electronegative phosphocholine head group fits into an electropositive pocket of the BC8 TCR formed by the amine backbones of the TCR α chain (Fig. 5d). This situation is reminiscent of the "cationic cup" that was recently described in the PG90 TCR, which mediates the phosphate unit of PG[24]. In both cases, the CDR loops that form the cationic cups sequester the anionic phospholipid antigen head group, either partially or completely for the BC8B and PG90

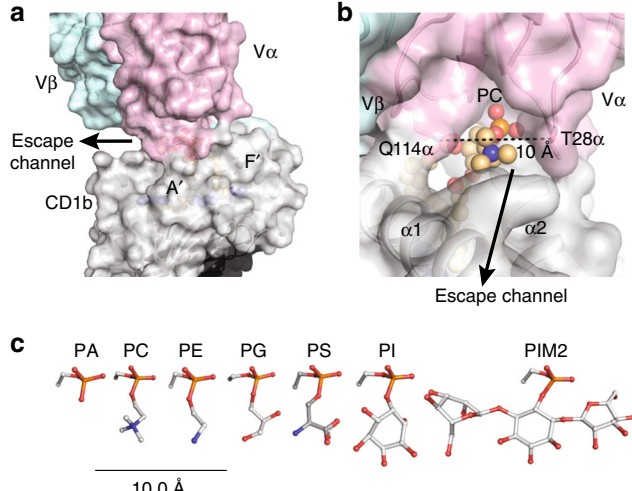

**Fig. 6** The lateral escape channel of the BC8B TCR. **a** BC8B docking onto CD1b creates a lateral escape channel above the A′ roof, to accommodate lipid antigen. The black arrow indicates the direction of the lateral channel opening. **b** The 10 Å escape channel is flanked by T28α and Q114α of the CDR1α and CDR3α loops, respectively. Distance measurement is indicated by the black dash. Color coding for BC8B, CD1b, and the PC antigen are as in Fig. 4a. **c** Chemical structures of the phospholipid head groups of phosphatidic acid (PA), phosphatidylethanolamine (PE), phosphatidylglycerol (PG), phosphatidylserine (PS), phosphatidylinositol (PI), and phosphatidylinositol dimannoside (PIM2) are shown in comparison to an atomic ruler

TCRs, respectively. To characterize the extent of antigen and TCR remodeling during BC8B TCR docking, we determined the crystal structures of the CD1b-PC complex to 1.7 Å and the unliganded BC8B TCR to 2.0 Å (Supplementary Table 1). Next these three structures were compared with the CD1b-PG–PG90 TCR, CD1b-PG and PG90 TCR[24], demonstrating distinct mechanisms of cationic cup formation (Fig. 5e, f). Comparison of CD1b-PG and CD1b-PG–PG90 TCR complexes shows that TCR binding resulted in extensive repositioning of the lipid, with the TCR β-chain moving 7 Å toward the PG head group and vice versa, pulling the antigen up slightly from its fully seated position in the cleft (Fig. 5e)[24]. Similarly, upon BC8B TCR docking, the PC antigen is lifted from the CD1b antigen-binding cleft in both the A′ and C′ pockets, allowing for significant TCR contacts with the sn1 acyl tail. The PC head group is lifted upwards by ~6 Å and bent, allowing for favorable contacts as the phosphate moiety is inserted into the BC8B TCR cationic cup (Fig. 5f).

Otherwise the formation mechanisms are highly distinct with regard to the cooperating TCR chains and the role of CD1b-phospholipid in driving formation. Whereas the BC8B TCR cationic cup is formed almost exclusively via the TCR α-chain (Fig. 5d–f), the PG90 TCR cationic cup was formed from interchain interactions among the CDR3α and CDR3β loops (Fig. 5a, e). For BC8B, the unliganded and CD1b-complexed BC8B TCR structures overlaid closely, with a root mean square deviation (R.M.S.D) of 0.65 Å. No significant movement of the CDR1α and CDR3α loops was observed between the BC8B TCR structures, revealing that the antigen interaction surface that contributes to the cationic cup is pre-formed (Fig. 5f). In the case of the PG90 TCR, the CDR3β loop undergoes a major conformational change upon CD1b-PG docking, demonstrating that the PG90 cationic cup is only formed after ligation (Fig. 5e). Thus, both autoantigen-reactive CD1b-restricted TCRs solved to data form a cationic cup, which appears to be a phosphate-selective feature within these TCRs. However, this structure can

be pre-formed within TCR-α or generated by inter-loop cooperation during binding of the negatively charged antigen.

**Cross-reactivity explained by head group escape channel**. Both PG90 and BC8B TCRs use cationic cups to position the phosphate at the base of the TCR. Whereas PG90 seals tightly over the phospholipid antigen head group[24], an escape channel forms between the CDR1α and CDR3α loops of the BC8B TCR. A lateral view shows that this escape channel is 10 Å at its maximum width between the side chains of T28α and Q114α of the CDR1α and CDR3α loops (Fig. 6a, b). This gap is large enough to accommodate the choline head group that defines PC (Fig. 6a, b). Although other glycerophospholipids were not co-crystalized with CD1b, their highly analogous structures (Fig. 2b) are consistent with a general mechanism in which their shared phosphoglycerol groups lie in the cationic cup. The 5 glycerophospholipids cross-reactively recognized by this TCR (PE, PG, PS, PI, PA) (Fig. 2f, g) would be expected to present ethanolamine, glycerol, serine, inositol, or no head group, respectively, towards the escape channel. Comparison of the molecular radii of these phospholipid head groups versus the size of the TCR channel predicts that all five could pass through the lateral gap in the TCR (Fig. 6c). Conversely, PIM2, which has a core PI structure that carries two additional mannose units, is predicted to be sterically hindered, and this ligand blocks CD1b tetramer binding (Fig. 2f, g). Last, this model plausibly explains why diversely loaded CD1b-endo tetramers stain the BC8B TCR. CD1b presents several abundant-cellular phospholipids, including PG[24], PC, PS, PE, and PI (Fig. 2a). If each of these lipids act as antigens via *trans*-TCR escape as proposed here, their combined occupancy of a population CD1b monomers increase the probability that two or more TCR binding CD1b proteins are present within any tetramer, as multimeric interactions are generally needed for T-cell staining.

## Discussion

In follow up to early evidence for phospholipid presentation by CD1d[37,38], these results provide new evidence for human T-cell response to CD1b and a broad range of common self-phospholipid autoantigens from unrelated human blood donors. From each of five donors tested, we describe a set of CD1b-restricted T-cells present in the blood that recognize ubiquitous self-lipids and are functionally autoreactive to CD1b-expressing APCs. Considering the identities of the antigenic lipids seen by these T-cell lines, they respond to phospholipids that form the majority of the biomass of every mammalian cell membrane. Further, we found that individual lines showed broad, cross-reactive recognition of many antigens[24,34,35], representing a new specificity pattern that is distinct from previous studies of CD1b-reactive T-cell clones that specifically recognize rare self-phospholipids associated with mitochondrial stress[20,24]. Whereas human B cell and immunoglobulin responses to phospholipids have been studied for decades as manifestations of autoimmune disease such as systemic lupus erythematosus[39,40], these data provide broader evidence for human for T-cell response to anionic phospholipids.

Further, the ternary TCR-PC–CD1b structure contributes a previously unknown mode of self-antigen recognition. The first autoreactive TCR footprint in the CD1a system bound to the A′-roof of CD1a itself rather than lipid antigen protruding from the F′ portal[29]. The first autoreactive TCR footprint in the CD1c system docked centrally over the CD1c antigen portal, which can occur only when a small ligand is present that is sequestered within CD1c and does not interfere with the TCR approach[41]. These two related mechanisms emphasize TCR binding to CD1

itself, where the carried lipid essentially does not contribute to the TCR epitope. This shared 'absence of interference' mechanism favors small lipids and can account for the promiscuous recognition of structurally diverse lipids, so long as the lipids do not protrude from the cleft to substantially cover the CD1 surface[42]. Here we describe a new mechanism for CD1b, where lipid head groups take a bent conformation, positioning the anionic phosphate into a cationic cup in the TCR. The more distal head group exits laterally through a 10 Å gap, which can act as a sizing mechanism. The TCR does not significantly interact with the parts of the head group that enter the escape channel, which explains its cross-reactivity to phosphatidic acid-based antigens with absent, small or large head groups. Larger head group, like the trisaccharide in PIM cannot likely cannot traverse escape channel based on its measured width. Sphingolipids, including sphingomyelin, which otherwise mimics the head group of PC, lack an anionic neck region for the cationic cup. This bent-neck, head group escape mechanism of BC8B contrasts with the only known TCR and CD1b–autoantigen interaction, where the PG90 TCR is highly specific for a phosphoglycerol head group[24].

In MHC–peptide–TCR structures, key interactions occur between the TCR, MHC and upward facing residues of the peptide bound within the MHC. These data contribute to the view that CD1–antigen–TCR interactions operate via the similar mechanisms. Indeed, CD1–antigen–TCR structures typically show that the distal, upward protruding head group of the lipid dominates antigen specificity. One counterexample is CD1b-presentation of GM1 by CD1b[27]. Here, additional sugar moieties could be added to a pentasaccharide core epitope without abrogating the T-cell response, leading to the speculation that the distal moieties might escape laterally below or within the TCR. Here we prove that the phospholipid neck region extends upward within the crevice between the TCR chains, makes specific interactions with a cationic cup and the head group exits laterally through an escape channel in the TCR. Because many types of head groups fit inside the channel and are ignored, the broad response to membrane phospholipids is mechanistically explained.

Early functional evidence indicates that certain aspects of the detailed mechanism observed for BC8B may be more broadly conserved. The TCR cationic cup mechanism of recognition of anionic phospholipids is seen in both CD1b–autoantigen–TCR complexes solved to date, though the cup forms by two clearly distinct mechanisms. The cationic cup observed here is pre-formed with the TCR α chain, whereas the PG90 cationic cup is formed by three variable loops in TCR α and β, and becomes a recognizable structure only after contact with CD1b-phospholipid[24]. Also, we isolated additional T-cell lines that show high specificity for small head groups (pattern 1) and lines with general phospholipid versus sphingolipid specificity (pattern 3) as examples of human T-cells recognizing common membrane phospholipids. CD1a, CD1b, and CD1d also bind and present membrane phospholipids, including phosphatidylcholines that are similar or identical to lipids studied here[28,41,43]. For CD1d, there is some evidence for promiscuous responses of clones to phospholipids[37], so plausibly use mechanisms similar to those described here.

Increasing evidence for promiscuous T-cell responses to cellular CD1 proteins and self-lipid antigens is beginning to raise questions about negative regulation of such responses. For CD1a and CD1c autoreactive T-cells can be directly detected in unfractionated blood[22,28], and our data suggest that CD1b-autoreactive cells can be readily detected with a simple selection procedure. The T-cell lines that we describe here, recognizing broadly distributed, abundant phospholipids were isolated from healthy blood bank donors. If such CD1b-specific T-cells are

generally present in human donors, the question arises as to whether they would cause autoreactivity or autoimmunity upon contact with any CD1b-expressing cell. Known mechanisms for limiting autoreactivity may be in play. T-cells may be suppressed by other cells or are inherently unable to perform inflammatory functions in vivo. They may not see the proper stimulus under steady-state conditions because the antigen, the antigen-presenting molecule, or costimulatory factors are in limited supply. Even though phospholipid antigens are widely available, CD1b expression in healthy tissues in the periphery is rare[44], but can be induced by Toll-like receptors, interleukin-1, cytokines, and other primary inflammatory signals[8,9,45,46]. Thus, in contrast to the ubiquitous expression of MHC class I proteins, activation and potentially harmful consequences of the T-cells described in this paper may be limited by tightly regulated CD1b expression in healthy tissues, and not by antigen availability.

On the other hand, such cells have the potential to cause autoimmunity, as was recently demonstrated in mice that expressed human CD1b and phospholipid specific TCR. These mice developed an autoimmune skin disease, when ApoE deletion causes marked overexpression of phospholipids in the skin[31]. This study raises the possibility that CD1b and phospholipids might play an unrecognized role in autoimmune disease. Also, clear evidence for immune response to phospholipids via CD1b and CD1d invites consideration of possible B cell–T-cell interactions in the anti-phospholipid antibody syndrome[47,48], an autoimmune state associated with systemic lupus erythematosus and other autoimmune diseases.

## Methods

**Tetramers and analytical flow cytometry.** CD1 monomers were obtained from the NIH tetramer facility. For loading of monomers, 32 µg of lipid was sonicated at 37 °C for 2 h in 90 µl of 0.5% CHAPS 50 mM sodium citrate buffer pH 7.4 in 10 mm diameter glass tubes. Subsequently, 20 µg CD1b monomer was added to the tubes and incubated overnight at 37 °C. Molecular Probes streptavidin-APC or streptavidin-PE was used for tetramerization. Human PBMC and T-cell lines were stained with tetramers at 2 µg ml$^{-1}$ in PBS containing 1% BSA and 0.01% sodium azide. Cells and tetramer were incubated for 10 min at room temperature, followed by addition of 3 µl of anti-CD3 (clone SK7-Fitc from BD biosciences) per 50 µl staining and another incubation for 10 min at room temperature and 20 min at 4 °C. Cells were analyzed using the BD LSRFortessa flow cytometer and FlowJo software.

**Lipids.** Phosphatidylcholine (PC, #850475; C18:1/C16:0), sphingomyelin (SM, #860584; C18:1/C16:0), phosphatidylinositol (PI, #840042; mixture from bovine liver with a range of fatty acids), phosphatidylserine (PS, #840032; mixture from porcine brain with a range of fatty acids), phosphatidic acid (PA, #840857; C18:1/C16:0), lyso-PA (#857130; C18:1), phosphatidylethanolamine (PE, #850757 C18:1/C16:0), phosphatidylglycerol (PG, #840503; C18:1/C18:0), and sulfatides (#131305P; mixture from porcine brain with a range of fatty acids) were purchased from Avanti polar lipids. Gangliosides GM1 (G7641; mixture from bovine brain with a range of fatty acids) and GM2 (G8397; mixture from bovine brain with a range of fatty acids) were purchased from Sigma. Phosphatidylinositol dimannoside (PIM) and diacyltrehalose (DAT), previously described as DAT2a[49], were provided by the Bill and Melinda Gates Foundation lipid bank. Cholesteryl 6-O-palmitoyl-β-galactopyranoside, also known as *Borrelia burgdorferi* glycolipid 1 (BBGL1) and 1,2-dioleyl-α-galactosyl-glycerol, also known as *Borrelia burgdorferi* glycolipid 2 (BBGL2) were synthesized as described previously[50,51]. *Salmonella* Typhi was cultured in Luria Broth. *Brucella melitensis* human isolate, strain X10017283-001 (CVI) was cultured in Tryptic Soy Broth. Lipid extracts were prepared by extracting bacterial pellet with choroform/methanol 1:2 (V:V) for 2 h at room temperature, followed by chloroform/methanol 2:1.

**CD1b–lipid complex analysis.** Human CD1b proteins made in U293 cells at the NIH tetramer facility as described[5,24] were transferred into a 15-ml glass tube and extracted with chloroform, methanol and water according to the Bligh and Dyer method[52]. The lipids were recovered from the organic layer and redissolved in chloroform/methanol (1:2) to a concentration equivalent to 5 µM of the input protein. The extracted lipids (5–10 µl) were loaded onto a nanospray tip for negative mode ESI−MS and multistage collision-induced dissociation tandem mass (CID–MS) using linear ion trap mass spectrometer (LXQ, Thermo Scientific). Collision energy was 20–35% of maximum and product ions were trapped with a *q* value of 0.25. The negative ions were calibrated with external reference Hexakis

(1H, 1H, 3H-tetrafluoropropoxy) phosphazine ([M+Cl]⁻ at $m/z$ 955.97 (Agilent part # I8720241)).

**Generation of T-cell lines**. PBMC were obtained from leukoreduction collars provided by the Brigham and Women's Hospital Specimen Bank, as approved by the Partners Healthcare IRB. Human PBMC and PBMC derived T-cells were stained with tetramers at 2 µg ml⁻¹ in PBS containing 1% BSA and 0.01% sodium azide. Cells and tetramer were incubated for 10 min at room temperature, followed by addition of 3 µl of anti-CD3 (clone SK7-Fitc from BD biosciences) per 50 µl staining and another incubation for 10 min at room temperature and 20 min at 4 °C. PBMC were sorted for positive staining with anti-CD3 and the indicated tetramers (Supplementary Fig. 1–5). Expansion of sorted cells was performed using anti-CD3 at 30 ng ml⁻¹ (clone OKT3), irradiated feeder cells, and IL-2. After 2 weeks, the sorting and expansion procedure was repeated once or multiple times. When the resulting cell lines consisted mainly of tetramer⁺ T-cells, they were screened for binding to a panel of antibodies against TCR Vβ and 2 µl of anti-CD4 (clone RPA-T4 from Biolegend) per 50 µl staining. Suitable combinations of antibodies were used to isolate subsets of the original T-cell lines (Supplementary Fig. 1-5). IFN-γ ELISPOT was performed using the 1D1K and GB-11-biotin antibodies (Mabtech), according to the manufacturer's instructions.

**TCR sequencing and transduction**. The TCR sequences were determined using RNA isolated with an RNeasy kit (Qiagen), and cDNA synthesized with a Quantitect reverse transcription kit (Qiagen). V segment usage was determined by PCR using primerset IPS000030 as described in www.imgt.org (Lefranc MP, 1989), as well as by a multiplex approach[53], followed by direct Sanger sequencing of the PCR product. Full length TCRα and TCRβ chains were cloned into a self-cleaving 2A peptide based (MSCV)-IRES-GFP (pMIG) vector[54] and co-transfected into HEK293T-cells in the presence of the retroviral packaging vectors pPAM-E and pVSV-g. The supernatant from the transfected HEK293T-cells was harvested and used to stably transduce TCR-deficient Jurkat clone 76 cells[55] to generate the BC13A T-cell line. After 5 days, cells that had the highest expression of GFP and CD3 were enriched by FACS sorting.

**Protein expression and purification**. Recombinant BC8B TCR were cloned into the pET30a vector and expressed, refolded and purified from *E. coli* inclusion bodies[24,56]. Inclusion bodies were resuspended in 8 M urea, 20 mM Tris–HCl (pH 8.0), 0.5 mM Na-EDTA, and 1 mM DTT. The TCR was refolded by flash dilution in a solution containing 5 M urea, 100 mM Tris (pH 8.0), 2 mM Na-EDTA, 400 mM L-arginine-HCl, 0.5 mM oxidized glutathione, 5 mM reduced glutathione, and EDTA-free anti-protease cocktail. The refolding solution was then dialyzed to eliminate urea. The resulting protein solution was then purified by size exclusion chromatography and HiTrap-Q anion exchange chromatography. Soluble CD1b was expressed from either insect *Trichoplusia ni* High Five cell lines, or HEK 293 S GnTI⁻ (American Type Culture Collection), and purified by HisTrap Ni²⁺-affinity chromatography and size exclusion chromatography[24,36]. For SPR and crystallography purposes, purified CD1b was loaded with a molar excess of target lipid in the presence of tyloxapol (Sigma) over a 16 h incubation period at 20 °C, then purified via anion exchange chromatography to homogeneity[24,36].

**Crystallization and structure determination**. Crystals of the BC8B TCR were grown via the hanging drop vapor diffusion method, using a protein-reservoir drop ratio of 1:1, at a protein concentration of 5 mg ml⁻¹ in 10 mM Tris–HCl (pH 8.0), with a crystallization condition of 20% (v/v) PEG 3350, 0.2 M Potassium sodium tartrate tetrahydrate. Crystals of the CD1b-PC–BC8B TCR ternary complex were grown at protein concentrations of 5 mg ml⁻¹, with a crystallization condition of 22% to 24% (v/v) PEG 3350, 0.01 M Tris–HCl pH 7.0–7.4. Crystals of the CD1b-PC complex were grown at protein concentrations between 5 and 8 mg ml⁻¹, with a crystallization condition of 22–26% PEG 3350, 0.2 M Sodium Iodide. Crystals of BC8B TCR, CD1b-PC–BC8B TCR, and CD1b-PC were soaked in a cryoprotectant comprised of reservoir solution containing 10% (v/v) ethylene glycol, before being flash-frozen in liquid nitrogen. Data were collected at the Australian Synchrotron at the MX2 beamline for the BC8B TCR and CD1b-PC–BC8B TCR crystals, and at the MX1 beamline for the CD1b-PC crystals[57]. Data were processed using the iMosflm software, and scaled using Aimless as part of the CCP4i program suite[58]. Crystal structures were solved via Molecular replacement using Phaser as part of the phenix program suite[59], with the structures of CD1b-PG (PDB accession code: 5WL1), and GEM42 TCR (PDB accession code: 4G8F) used as models for solving CD1b and TCR structures, respectively. Manual adjustments of the models were conducted in the coot graphics program[60], following maximum-likelihood refinement with Buster 2.10[61]. Electron density for ligands were well defined in the CD1b-PC–BC8B TCR structure. Electron density for regions of the PC molecule in the CD1b-PC structure were less defined, indicative of flexibility. All molecular representations were generated in PyMOL. BSA values were calculated using areaimol, and contacts generated using the CONTACT program in the CCP4i program suite[58].

**Surface plasmon resonance**. SPR analysis on the BC8B TCR against CD1b-phospholipids was conducted on the BIAcore 3000 instrument at 25 °C in 10 mM Tris–HCl (pH 8.0), 150 mM NaCl, and 1% (w/v) bovine serum albumin[24]. Binding of the recombinant, refolded TCR in solution to CD1b, loaded with exogenous lipids and amine coupled to a CM5 chip, was analyzed. All experiments were conducted as two or three independent experiments in duplicate. Data analysis and visualization were generated using Graphpad Prism 7.0, using the 1:1 Langmuir binding model.

**Reporting summary**. Further information on experimental design is available in the Nature Research Reporting Summary linked to this article.

## Data availability
Structural data were deposited in the Protein Data Bank, with the following accession codes: BC8B TCR (6CUH), CD1b-PC–BC8B (6CUG), and CD1b-PC (6D64). All remaining data are available within the article and its supplementary information files and from the corresponding authors on request. A reporting summary for this article is available as a Supplementary Information file.

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

## Acknowledgements

This work was supported by the Australian Research Council and National Health and Medical Research Council, the National Institutes of Health (AR048632 and AI049313), the Bill and Melinda Gates Foundation, and Netherlands Organization for Scientific Research (NWO). S.G. is a Monash Senior Research Fellow. J.R. is supported by an ARC Laureate Fellowship.

## Author contributions

A.S., P.R, J.F.R., and S.L. undertook the research and analyzed the data. A.S., P.R., S.G., T.-Y.C., D.B.M., I.V.R., and J.R. designed the research and analyzed the data. M.H., A.J.M., J.A., J.P., J.K.-K. provided unique reagents. D.B.M., J.R., I.V.R. wrote the manuscript.

## Additional information

**Competing interests:** The authors declare no competing interests.

