## [Peer Review File · Nature Communications]

Reviewers' comments:

Reviewer #1 (Remarks to the Author):

The manuscript by Rhijn and Rossjohn describes the identification, isolation and biochemical and characterization of several CD1b-restricted T cell clones that recognize self phospholipids but for the most part not sphingolipids. Using crystallography they further identify the structural basis of how a variety of different phospholipids are recognized but the same TCR, clone BC8B, even though they carry different chemical headgroups. The BC8B TCR specifically makes contacts with the central phosphate moiety but forms an escape channel that allows the different headgroups to laterally escape from the CD1b-TCR interface over the A' pocket of CD1b.

The work is technically sound, of importance to the general immunology audience and fills gap in our current knowledge of self-phospholipid recognition by CD1b-restricted T cells. The work is also presented very clearly and the results are interpreted and discussed appropriately.

I only have one minor concern that the authors should address before publication.

While the authors state that the SPR and Tetramer staining data correlate very well, this is not true for the lipid PS. CD1b-PS tetramers do not stain BC8B T cells (Figure 2F), while CD1b-PC does stain with a high MFI. However, in the SPR experiment both lipids exhibit similar affinity for the TCR (15 versus 11 μ M). The authors should address this disconnect.

Dirk Zajonc

Reviewer #2 (Remarks to the Author):

This study builds upon previous work investigating TCR interactions with CD1b presenting endogenous ligands, primarily glycerophospholipids. The authors use CD1b tetramers generated previously to select T cell line specific for bacterially derived ligands and find crossreactivity with endogenous self-ligands presented by CD1b. An observation made previously. Three broad patterns of CD1b/lipid binding are described including: TCRs that are highly specific for phospholipids bearing small solvent exposed polar moieties, TCRs that exhibit promiscuous binding to both phospholipids and sphingolipids and TCRs that, despite similar polar head groups, recognize only phospholipids and not sphingolipids.

This finding is explained in depth by crystallization of a representative TCR, BC8B, that exhibits novel recognition of the phosphate ester linkage and not the amide linkage present in sphingolipids. The authors further demonstrate interaction of the TCR results in a channel along the TCR CD1b interface, in which the polar moiety can lie enabling TCR interaction with the lipid 'neck region'. A comparison of structural binding features with other CD1b restricted TCRs merely shows different TCRs with the same restriction, share some features of interaction while differing in their binding to specific antigens, which is not unexpected. The novel observation that the T cell clone BC8B binds to the 'neck region' of glycerophospholipids while ignoring the exposed head group however, is noteworthy. Indeed, the relevance of the cationic cup that is extensively commented on extends only as far as the initial selection of TCRs that bind to anionic lipid moieties.

Overall, this study provides a novel mode of recognition of lipid ligands. Little consideration is given to the wider relevance of the cells and focus is maintained on structural aspects of interaction compared previously solved structures. Given these are novel TCRs, is it not expected that different reactivity and binding interactions occur?

The manuscript is well written, the evidence is presented clearly and the supplemental data is justified. There are a few comments to address however:

1. Do these TCR react to APC expressing physiological levels of CD1b?
2. Tetramer staining and selection showed abundance of T cells specific for common membrane lipids. Can the authors comment on the frequencies of these T cells and their phenotypic characteristics?
3. Mass spec. of endogenous ligands found within CD1b loaded is performed. The authors might comment on the relative abundance of different endogenous lipids found in secreted CD1b and compare with the endogenous lipids found bound to other CD1 molecules.
4. Which are the average acyl chain lengths of CD1b-eluted lipids?
5. Despite the fact these TCRs were derived with bacterial ligands there is no mention later of their specificity towards these lipids. This observation suggests broader lipid recognition than with just glycerophospholipids and sphingolipids. Could the authors comment on proposed mechanism of interaction with these ligands?
6. Line 128. Shamshiev et al. (*Immunity*, 13, 255, 2000) reported the first comparison of constraints provided by hydrophilic head of self-glycolipids. While GM1 and other glycosphingolipids with bigger head were immunogenic, self-glycosphingolipids with smaller head were not stimulatory. This study considered relevant the distal Galactose in GM1, that might resemble the recognition of the phosphate neck described in this manuscript. The authors may comment on this part.
7. Which is the relative abundance of the three patterns of CD1b-autoreactive T cells?

Minor points:

Line 127. The first identified self-lipid antigens presented by CD1b were glycosphingolipids and were reported in 1999 (Shamshiev et al., *Eur.J.immunol.*).

Line 137. The authors should clarify whether they use T cell clones or T cell lines and should refer to them in a consistent manner.

Line 163, T cells are "restricted to CD1b" and not "specific for CD1b"

Line 396. Indicate the acyl chain length of each used lipid.

Reviewer #1 (Remarks to the Author):

We thank the reviewer for careful evaluation of our paper and for stating: “The work is also presented very clearly, and the results are interpreted and discussed appropriately.” The reviewer raises one minor concern:

“While the authors state that the SPR and Tetramer staining data correlate very well, this is not true for the lipid PS. CD1b-PS tetramers do not stain BC8B T cells (Figure 2F), while CD1b-PC does stain with a high MFI. However, in the SPR experiment both lipids exhibit similar affinity for the TCR (15 versus 11 μM). The authors should address this disconnect.”

Thank you for this astute observation. This comment arises in part due to oversimplified labeling in which we grouped FACS profiles together with one label in the middle panel of 2F. In fact, among the four histograms on the left, CD1b-phosphatidylserine is low, not absent. We revised to correct this error by separately and more clearly labeling the 4 tetramers in this group in 2F.

Yet after this change, the reviewer's comment remains correct in that the tetramer assay shows a higher separation between CD1b-PS and CD1b-PC than that observed by our SPR analyses. In response, we carried out further tetramer and SPR assays. After loading of a new set of tetramers and using them to stain BC8B cells, we found that CD1b-PS stained somewhat higher this time (see figure below), while other tetramers showed essentially the same results. We have also repeated the SPR results and confirm that the affinity values obtained in the second experiment (15.3 μM) match the value that reported in the initial submission (15.3 μM). Although the precise basis of differences between tetramer staining and SPR is not fully resolved, the two assays have differing loading protocols and different stoichiometry, and we consider the SPR to be more quantitative.

However, the most important point, that the overall pattern of BC8 selectively responding to phospholipids over sphingolipids, is clear in both tetramer and SPR assays. Also, both techniques show that CD1b-phosphatidylcholine is greater than phosphatidylserine. We also revised (page 7) to acknowledge the reviewer's point:

“Although there were some differences in avidity or affinity among individual phospholipids, both tetramer and SPR assays showed a clear preference for self-phospholipids over sphingolipids and promiscuity among phospholipids, implying neck region specificity”.

Reviewer #2 (Remarks to the Author):

We thank the reviewer for their critique of our paper and for stating that the ‘neck’ region binding observation is “noteworthy” and that “Overall, this study provides a novel mode of recognition of lipid ligands.” The reviewer makes a number of points that we address in turn below.

Little consideration is given to the wider relevance of the cells and focus is maintained on structural aspects of interaction compared previously solved structures. Given these are novel TCRs, is it not expected that different reactivity and binding interactions occur?

This is the only the third ternary structure solved for CD1b, so new structures in an emerging field are expected to yield new interactions and this one does. However, the key structural conclusions are not considered new simply because known concepts are seen here in CD1b. Instead we highlight certain concepts not previously seen for any MHC or CD1 protein. Also, the TCR escape channel has explanatory power for broad responses to phospholipids over sphingolipids.

The final two paragraphs of the discussion consider the wider biological relevance of these cells to possible autoimmunity and candidate mechanisms of their control. Extending this overview, we accept the reviewer's challenge to more clearly highlight the overarching structural issues and more extensively cover prior functional studies such as the core epitope concept. On page 13, we added:

"In MHC-peptide-TCR structures, key interactions occur between the TCR, MHC and upward facing residues of the peptide bound within the MHC. These data contribute to the view that CD1-antigen-TCR interactions operate via the similar mechanisms. Indeed, CD1-antigen-TCR structures typically show that the distal, upward protruding head group of the lipid dominates antigen specificity. One counterexample is CD1b-presentation of GM1 by CD1b²⁷. Here, additional sugar moieties could be added to a pentasaccharide core epitope without abrogating the T-cell response, leading to the speculation that the distal moieties might escape laterally below or within the TCR. Here we prove that the phospholipid neck region extends upward within the crevice between the TCR chains, makes specific interactions with a cationic cup and the head group exits laterally through an escape channel in the TCR. Because many types of head groups fit inside the channel and are ignored, the broad response to membrane phospholipids is mechanistically explained."

The manuscript is well written, the evidence is presented clearly and the supplemental data is justified.

1. Do these TCR react to APC expressing physiological levels of CD1b?

Yes. Although not reported in the initial submission, we also used monocyte derived dendritic cells, which are the best available primary cell that reflects human in vivo function of CD1b. However, because they also express CD1a, CD1c, and CD1d, DCs do not allow direct identification of the restricting antigen presenting molecule. The artificial antigen presenting cell lines used here to establish CD1b response (C1R, K562) generally match CD1 expression levels in human DCs, and in many published studies we have used these three cell types interchangeably. If anything, DCs tend to give higher responses (see Fig. 3 of Moody, NI, 2002). However, tumor cells have other limitations, so it is fair to ask to see T cell responses directly from a native APC. We now show responses against CD1b+ DC with additional experiments and report them in revision (see page 4 and Supplementary Fig. 6),

which is copied here.

2. Tetramer staining and selection showed abundance of T cells specific for common membrane lipids. Can the authors comment on the frequencies of these T cells and their phenotypic characteristics?

Frequencies of these CD1b-endo tetramer+ cells among polyclonal T cells are shown in the upper left panels of Supplementary Figs 1-5. In revision, we noticed that the percentage was missing in Supplementary Fig. 1 and include it now. Also, as requested we now comment specifically on the precursor frequencies on page 3. Frequencies of these CD1b tetramer+ cells are: 0.063%, 0.028%, 0.33% (though this particular gate was not stringent enough to exclude all tetramer-negative events), 0.054%, 0.018%. We added discussion pointing out that these numbers are similar among donors, higher than that for naive MHC-restricted clones and somewhat lower than human NKT cells. With regard to phenotype, we expect that the reviewer is asking about effector function to which we can say that they make interferon- γ . However, detailed effector analysis of lines grown moderately long term in vitro is not reliable, and direct ex vivo phenotyping with high purity sorting is extremely challenging. We do emphasize "phenotype" with regard to the many types of phospholipid autoantigens recognized, which is the aspect of the T cell data most directly connected to the structural results.

3. Mass spec of endogenous ligands found within CD1b loaded is performed. The authors might comment on the relative abundance of different endogenous lipids found in secreted CD1b and compare with the endogenous lipids found bound to other CD1 molecules.

Prompted by the reviewer, we added discussion to say that CD1-self lipid complexes can occur in other CD1 systems on page 13, while being careful to highlight that structurally solved TCR escape channels are unknown for other CD1 isoforms.

"CD1a, CD1b and CD1d also bind and present membrane phospholipids, including phosphatidylcholines that are similar or identical to lipids studied here^{28,30,42}. For CD1d, there is some evidence for promiscuous responses of clones to phospholipids³⁷, so plausibly use mechanisms similar to those described here."

4. Which are the average acyl chain lengths of CD1b-eluted lipids?

We have deduced the acyl chain lengths of phosphatidylcholine, phosphatidylethanolamine, phosphatidylinositol, phosphatidic acid and sphingomyelin and added them to the nanospray MS data in Fig 2a. Chain length varies from 34 to 42 with strongest signals seen for C36-C40 lipids. Using HPLC-TOF-MS, we show the combined chain length for one representative family of phospholipid ligands (PC) and one sphingolipid family (sphingomyelin), totaling 40 ligands (Supplementary Fig. 8). These profiles match those seen in nanospray MS with a range of C30 to C42 and median values of C38 and C40.

5. Despite the fact these TCRs were derived with bacterial ligands, there is no mention later of their specificity towards these lipids. This observation suggests broader lipid recognition than with just glycerophospholipids and sphingolipids. Could the authors comment on proposed mechanism of interaction with these ligands?

In the initial submission we show details of the T cell recognition of bacterial extracts or synthetic bacterial molecules in Supplementary Figs. 1-5. Just as the reviewer says, we did not discuss these in the main manuscript in detail due to *Nature Communications* length limits. These are somewhat complex patterns which are separate from the main theme of the manuscript. The key antigens used and self or foreign antigens recognized are summarized in tabular form in Table 1. As requested, in revision we added a proposed mechanism of how foreign lipid treated tetramers could generate autoreactive T cells on page 4:

“Although somewhat counterintuitive, the recovery CD1b autoreactive T-cells has been observed previously²⁰ and could be explained by extremely cross-reactive TCRs that do not discriminate lipid structure, retention of self-lipids from the expression system used to make CD1b tetramers, or antigenic lipids expressed both by mammalian cells and bacteria.”

6. Line 128. Shamshiev et al. (Immunity, 13, 255, 2000) reported the first comparison of constraints provided by hydrophilic head of self-glycolipids. While GM1 and other glycosphingolipids with bigger head were immunogenic, self-glycosphingolipids with smaller head were not stimulatory. This study considered relevant the distal Galactose in GM1, that might resemble the recognition of the phosphate neck described in this manuscript. The authors may comment on this part.

Good point. This early paper used processing-independent recognition studies, and it has certain parallels with the current work, including cross-reactive recognition of sphingolipids in which structures can be added to a core epitope. However, unlike the current report, the positioning of the epitope between CD1b and the TCR was not established structurally. Also, we remain a bit skeptical of direct parallels, as the five-carbohydrate core is much larger than the phospholipid neck region. We have added discussion of Shamshiev's work on page 13, which is copied here:

“One counterexample is CD1b-presentation of GM1 by CD1b²⁷. Additional sugar moieties could be added to a pentasaccharide core epitope without abrogating T-cell response, leading to the speculation that the distal moieties might escape laterally below or within the TCR. Here we prove that the phospholipid neck region extends upward within the crevice between the TCR chains, makes specific interactions with a cationic cup and the head group exits laterally through an escape channel in the TCR. Because many types of head groups fit inside the channel and are ignored, the broad response to membrane phospholipids is mechanistically explained.”

7. Which is the relative abundance of the three patterns of CD1b-autoreactive T cells?

To directly answer the reviewer's question as framed, we discussed amongst ourselves whether it is feasible to measure the three patterns in human cohort studies. Measuring the three patterns ex vivo in a cohort is probably not feasible. Here we used 13 different tetramers to establish one pattern and repeated the process for five donors. To really answer the question, we would require a population of donors and quantitative measurements with 13-color tetramer panel, which is currently technically not possible.

The absolute precursor frequency of these tetramer-positive T cells in 5 human donors (which somewhat approximates the question asked) is outlined under point 2 above. Also, considering both the phosphatidylglycerol specific T cells seen previously (Van Rhijn, PNAS, 2016) and the common membrane phospholipid reactive T cells seen here, we can say such CD1b and phospholipid autoreactive cells are detected in nearly all donors tested. Therefore, CD1b and phospholipid reactive T cells appears represents a previously unrecognized but prevalent human T cell type.

Minor points:

Line 127. The first identified self-lipid antigens presented by CD1b were glycosphingolipids and were reported in 1999 (Shamshiev et al., Eur.J.immunol.).

This key paper on CD1b autoantigens is discussed on page 5. The manuscript now reads:

“Two decades after the discovery of neural tissue-derived glycosphingolipid presentation by CD1b³³, ubiquitous antigen-presenting cell-intrinsic CD1b-presented autoantigens have only been recently discovered^{20,31}.”

Line 137. The authors should clarify whether they use T cell clones or T cell lines and should refer to them in a consistent manner.

We had internal discussion on this point, and for accuracy, assigned the term clone only when rigorously proven. As outlined in Supplementary Figs. 1-5, T cell populations were sorted until we could gate monomorphic populations in flow cytometry, which suggests but does not prove clonality. Even when we could only obtain a single TCR sequence from these lines, formally they have not gone through a single cell 'cloning' procedure. We use “clone” to refer only to previously published clones and to Jurkat cells transduced to carry a single TCR.

Line 163, T cells are “restricted to CD1b” and not “specific for CD1b”

Throughout the paper we use “CD1b-restricted” (lines 61, 295, 323, 656), but in line 163 “Despite this extreme promiscuity for lipid structure, both lines were still specific for the CD1b isoform in T cell activation” we wanted to keep the possibility open that CD1b itself is the antigen, and that the lipid that is bound is irrelevant.

Line 396. Indicate the acyl chain length of each used lipid.

We have included the chain lengths of the purchased lipids and the naturally included lipids in the relevant materials and methods section on page 15.

REVIEWERS' COMMENTS:

Reviewer #1 (Remarks to the Author):

All my concerns have been appropriately addressed!

Reviewer #2 (Remarks to the Author):

The revised manuscript is clearly written and the authors have demonstrated careful consideration of the comments. Relevant information has been added to the manuscript, of prominence, evidence of T cell activation toward CD1b expressed on monocyte derived dendritic cells. T cells that recognize self-lipids presented by CD1b are seemingly a distinct compartment that may have important roles in diverse pathologies that warrants further investigation, but out of the scope of this manuscript.